# Counterfactual Neural Temporal Point Process for Estimating Causal Influence of Misinformation on Social Media

**Yizhou Zhang**\*, **Defu Cao**\*, **Yan Liu**
Department of Computer Science
Viterbi School of Engineering
University of Southern California
{zhangyiz,defucao,yanliu.cs}@usc.edu

## Abstract

Recent years have witnessed the rise of misinformation campaigns that spread specific narratives on social media to manipulate public opinions on different areas, such as politics and healthcare. Consequently, an effective and efficient automatic methodology to estimate the influence of the misinformation on user beliefs and activities is needed. However, existing works on misinformation impact estimation either rely on small-scale psychological experiments or can only discover the correlation between user behaviour and misinformation. To address these issues, in this paper, we build up a causal framework that model the causal effect of misinformation from the perspective of temporal point process. To adapt the large-scale data, we design an efficient yet precise way to estimate the **Individual Treatment Effect** (ITE) via neural temporal point process and gaussian mixture models. Extensive experiments on synthetic dataset verify the effectiveness and efficiency of our model. We further apply our model on a real-world dataset of social media posts and engagements about COVID-19 vaccines. The experimental results indicate that our model recognized identifiable causal effect of misinformation that hurts people's subjective emotions toward the vaccines.

## 1   Introduction

Recent researches reveals that widespread fake news and misleading information have been exploited by misinformation campaigns to manipulate public opinions in different areas, such as healthcare [36, 38, 18] and politics [20]. To address this crucial challenge, research efforts from different perspectives have been devoted, such as fake news detection and coordination detection [35, 36, 38].

However, an essential associated research question has not been explored sufficiently: how to know a piece of misinformation's causal influence on a user's beliefs and activities on a large-scale social media. Precisely estimating such impact is crucial for misinformation mitigation in various areas, e.g. delivering the corresponding clarification contents to the users that are most likely to be affected, allocating resources for more efficient and effective misinformation mitigation, and helping researchers understand misinformation campaigns better. Nevertheless, most of existing researches in social media analysis focus on understanding correlation between misinformation and user activities, rather than causal effect [26, 38, 18, 45]. As a result, they can not distinguish the effect from personal prior beliefs and engagement with misinformation. Current researches on misinformation's causal influence on people are mainly from psychology field [15, 42]. They are usually based on carefully designed psychological randomised controlled trials on recruited subjects. Thus, it is impossible to

---

\*Equal Contribution

36th Conference on Neural Information Processing Systems (NeurIPS 2022).

extend them onto large-scale social media platforms due to the high cost to recruit enough subjects and ethical risk in conducting such a large-scale psychology experiments.

Since personal beliefs are usually unobservable, researchers usually apply the feature of the tweets generated or retweeted by the users as a proxy [38]. However, the lack of appropriate algorithmic tools to conduct causal analysis on social media posts prevents researchers from understanding causal effect of misinformation. The processes that social media users generate original posts and engage with existing posts are typical temporal point processes. But existing methodologies for temporal causal effect estimation mostly focus on covariates and outcomes continuously distributed on timeline [4, 3, 2, 6, 16], rather than discrete event points randomly scattered on timeline. Although the essential theory for counterfactual analysis of point process is already established[30], most works are motivated by healthcare and thus focus on the hazard models, e.g. survival analysis [1] or the chance to catch cancer [31], which only consider the single occurrence of the most recent future event. However, on social media, we care more about multiple events happening in a time window. [14] and [24] are rare works studying the causal effect on multiple occurrences in temporal point process. But [24] mainly focus on simulating the counterfactual events given an intervened intensity function rather than learning the treatment effect of specific factors on the process. As for [14], one of its assumption is that the event marks must be categorized to a finite number of classes, leaving no space for the rich continuous features of social media posts, such as user sentiment and subjectivity scores.

In this work, we propose a framework that models the causal effect of a given piece of information on user beliefs and activities via counterfactual analysis on temporal point process [37, 12, 22, 48, 51, 25, 7] with continuous features. We first define a causal structure model that characterizes the misinformation impact as how the engagement with the misinformation change a user's intensity function of generating original posts. In this model, the engagement with misinformation is considered as the treatment, and the user's future conditional intensity function is considered as the outcome. Then we design a functional that converts the change of two functions to a vector with intuitive physical meaning [23]. To estimate the effect, we design a neural temporal point process model. It disentangles the distribution of event timestamp and post feature (e.g. the text embedding). Then it models the distribution of post features and event timestamp with Gaussian Mixture Model and temporal point process respectively. Such design enables it to acquire a closed-form solution of the feature expect without losing expressive power, leading to a balance between precision and efficiency.

A critical challenge in training neural networks to recognize causal effect is the hidden bias in the dataset. In social media data, the most crucial bias is from information cocoons [50]: users tend more to engage with the contents that they are interested in, and thus personalized recommendation systems will deliver user more contents that they are interested in to increase user engagement. Such bias leads to a data distribution different from randomised controlled trials and thus make neural networks give biased estimation. To decorrelate time-varying treatment from user's covariates and history in point process, we apply adversarial training to optimize a min-max game. More specifically, the encoder tries to minimize the likelihood of the observed treatments while a treatment predictor tries to maximize it. Our theoretic analysis proves that any balanced solution of the min-max game, rather than the global optimal solution in existing works [4], can help us remove the bias from information cocoons. In addition, the extensive experiments on synthetic datasets and real-world datasets indicate that our framework is able to approximate unbiased and identifiable estimation on the causal effect. In conclusion, the main contributions of the proposed model are as follows:

- We propose a novel research problem on misinformation impact, which aims to find the causal effect of misinformation on users' belief and activities on social media.
- We propose a causal structure model to quantify the causal effect of misinformation and further design a neural temporal process model to conduct unbiased estimation to the effect.
- We evaluate our model on synthetic datasets to examine its effectiveness and efficiency and ues it to recognize identifiable causal effect of misinformation from real-world data.

## 2 Related Work

### 2.1 Influence of Misinformation

Recent researches about misinformation mainly focus on detecting fake news[35, 27, 32, 34], misinformation campaign detection [37, 49] and understanding how fake news attract user engagement[8, 9].

Some researcher attempts to study the relation between misinformation and people's behaviours [15, 42, 38, 18]. However, most of them focus on mining the correlation between misinformation and people behaviours rather than causal effects. Only a limited amount of works, such as [42] try to understand the causal effect. However, they are usually from psychology field, and mainly rely on carefully designed randomised controlled trials. Extending such trials on large-scale social media platforms brings not only high cost but also potential ethical risk.

## 2.2 Temporal Point Process

The process that a user retweet or posts tweets can usually be modeled as a temporal point process with event feature [37, 52, 33, 10]. A temporal point process (TPP) with event feature is a stochastic process whose realization is a sequence of discrete events in a continuous timeline: $S = [(\boldsymbol{f}_1, t_1), (\boldsymbol{f}_2, t_2), ...]$, where $\boldsymbol{f}$[2] is the event feature (a scalar or a vector) and $t$ is the timestamp of the event. A TPP is fully characterized by an intensity function $\lambda(\boldsymbol{f}, t|S_h)$ defined in the following integral equation:

$$\mathbb{E}(N(\boldsymbol{F}, T_1, T_2)|S_h) = \int_{\boldsymbol{F}} d\boldsymbol{f} \int_{T_1}^{T_2} \lambda(\boldsymbol{f}, t|S_h)dt \tag{1}$$

where $\boldsymbol{F}$ is an area in the feature space, $S_h$ is the historical sequence of all events happening before time $T_1$, $N(\boldsymbol{F}, T_1, T_2)$ is the number of events whose feature vectors are in $\boldsymbol{F}$ and timestamps are in the range $[T_1, T_2]$. The meaning of $\lambda(\boldsymbol{f}, t|S_h)$ is the expected instantaneous speed that the user generate posts at point $\boldsymbol{f}$ in the feature space on time $t$. The process after time $t_i$ is fully described by $\lambda(\cdot, \cdot|S_h)$ [7]. Recent works propose to apply neural networks to model the $\lambda$ function [12, 22, 48, 51, 25, 7].

## 2.3 Counterfactual Analysis on Temporal Point Process and Continuous Time Series

The works focusing on studying the causal effect on multiple occurrences in temporal point process are rare. In [24], the authors mainly focus on the sampling of counterfactual events rather than the learning the influence of specific factors on the intensity function. Another work[14] proposes a counterfactual analysis framework to understand the causal influence of event pairs in temporal point process. It defines the individual treatment effect (ITE) of an event toward future process as:

$$ITE = \mu_y^1(t, t + T) - \mu_y^0(t, t + T) = \frac{1}{T}\int_t^{t+T} \lambda_y^1(t) - \lambda_y^0(t)dt \tag{2}$$

where $\mu_y$ is the expect of the event number of type $y$ per unit time in time range $[t, t+T]$. $\mu_y^1$ indicate the case that a treatment is applied (exposed to misinformation) and $\mu_y^0$ is in contrary. However, this metric is only suitable in the case that the events can be categorized to finite discrete types. This is not applicable for social media post because most meaningful features of the posts, such as geographical information, sentiment score and subjective score, are naturally continuous. Forcibly discretizing them will lose meaningful information. Besides, for counterfactual analysis of time series, [4] proposes CRN, a neural model that can learn unbiased estimation to counterfactual world and causal effect. [3] proposes to analyze counterfactual estimation using synthetic controls via a novel neural controlled differential equation model. [2] introduces a new causal prior graph to avoid the undesirable explanations that include confounding or noise and use a multivariate Gaussian distribution to model the real continuous values. However, all of them focus on modeling an observable variable existing on a continuous timeline instead of temporal point process. Unless getting heavily revised, such previous work can not be simply transferred to our problem scenario.

# 3 Proposed Causal Structure Model and Treatment Effect

## 3.1 Causal Structure Model

In this study, we focus on understanding how a user's engagement with the misinformation post causally influence the characters of the posts generated by the user in a fixed future time window. We formulate the process that a user interact the post shared by others and generate social media posts as

---

[2]We use bold font to emphasize that $\boldsymbol{f}$ is a feature vector rather than a function.

two temporal point process where each event carries a continuous outcome vector. We denote the process of engaging the diffusion of a post as $P_e$ and the process generating new posts as $P_g$. Then the realization of the two temporal point process are respectively two sequences $S_e$ and $S_g$ of discrete events with continuous outcome vector in a continuous time range:

$$S_e = [(\boldsymbol{f}_1^{(e)}, t_1^{(e)}), (\boldsymbol{f}_2^{(e)}, t_2^{(e)}), ...], \quad S_g = [(\boldsymbol{f}_1^{(g)}, t_1^{(g)}), (\boldsymbol{f}_2^{(g)}, t_2^{(g)}), ...] \tag{3}$$

where $\boldsymbol{f}$ is the feature vector char-acterizing the event and $t$ is the time stamp. For $S_e$, each event corre-spond to an interaction (e.g. "like" or comment), and the vector $\boldsymbol{f}^{(e)}$ is the feature of the content (like the text representation, sentiment score and metadata) from others. Simi-larly, for $S_g$, the vector $\boldsymbol{f}^{(g)}$ is the feature of the content generated by the user. To examine the causal ef-fect of an interaction event on the posts generated by the user in the future, we construct the following

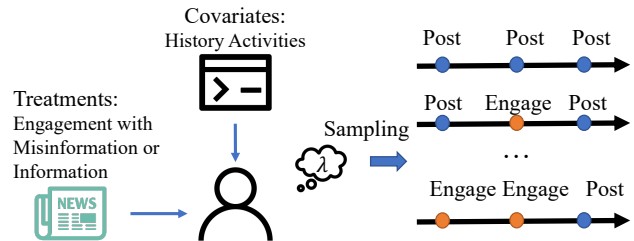

Figure 1: The proposed causal structured model describing the impact of a piece of information on user.

causal structure model, formulated as $< X, Y, Tr >$, where $X$ is the covariate, $Y$ is the outcome and $Tr$ is the treatment. In this model, given an interaction event $(\boldsymbol{f}_i^{(e)}, t_i^{(e)})$ whose causal effect is to be examined, we consider the this event as the treatment $Tr$, and all the events, including both engagement events and posting events, that happen before $t_i^{(e)}$ are considered as the covariates. As for the outcome, rather than simply considering the most next generation event after $t_i^{(e)}$, we need a representation that can reflect the change of the whole generating process $S_g$ in a fixed future time window $T$. Thus we apply the conditional intensity function of the future process, denoted as $\lambda(\boldsymbol{f}, t | Tr \cup X)$, as outcome. As discussed in the related work section, $\lambda$ function completely describe the future process. The overview of the model is presented in Figure. 1.

## 3.2 Treatment Effect Evaluation

In traditional counterfactual analysis works, the outcome is usually a scalar or a vector with finite dimensions. Thus, the treatment effect can be trivially computed by comparing the difference of the outcomes from real world and counterfactual world. However, in our framework, it is non-trivial to compute the difference of two functions. To overcome this challenge, we propose to first apply a functional $\mathcal{F}$ to project the $\lambda$ function to a vector with finite dimensions:

$$\mathcal{F}_T(\lambda, Tr \cup X) = \frac{\phi(t, t+T, \lambda, Tr \cup X)}{\mu(t, t+T, \lambda, Tr \cup X)} \tag{4}$$

$$\phi(t, t+T, \lambda, Tr \cup X) = \mathbb{E}_{S \sim P(S | \lambda, Tr \cup X)}[\sum_{(\boldsymbol{f}_i, t_i) \in S_{t:t+T}} \boldsymbol{f}_i] \tag{5}$$

$$\mu(t, t+T, \lambda, Tr \cup X) = \mathbb{E}_{S \sim P(S | \lambda, Tr \cup X)} |S_{t:t+T}| = \mathbb{E}(N(\sup(\boldsymbol{f}), t, t+T) | Tr \cup X) \tag{6}$$

where $P(S | \lambda, Tr \cup X)$ is the distribution of the event sequence $S$ sampled from the temporal point process described by $\lambda(\cdot, \cdot | Tr \cup X)$, $T$ is the time window that is a hyper-parameter, $\sup(\boldsymbol{f})$ is the support set of $\boldsymbol{f}$ (the area where the probability density is larger than 0), and $S_{t:t+T}$ is a sub-sequence of $S$. $S_{t:t+T}$ contains every event in $S$ that happens at a time between $t$ and $t+T$. The intuitive meaning of $\mathcal{F}$ is the expected mean feature vector of all posts generated by a user. Thus, by comparing the outputs of $\mathcal{F}$ in real world and counterfactual world, we can see how the engagement with a specific post change the average features, e.g. general sentiment scores or text embedding. With this functional, we can simply compute the individual treatment effect as:

$$ITE = \mathcal{F}_T(\lambda, Tr \cup X) - \mathcal{F}_T(\lambda, \emptyset \cup X) \tag{7}$$

where $\emptyset$ is an empty set, $\mathcal{F}_T(\lambda, \emptyset \cup X)$ is the functional from counterfactual world in which we assume that the treatment is not applied (e.g. the misinformation post is not recommended or labeled as

misinformation). For brief, we write $\mathcal{F}_T(\lambda, \emptyset \cup X)$ as $\mathcal{F}_T(\lambda, X)$ The overall impact of the treatment can be represented with the average treatment effect:

$$ATE = \mathbb{E}_{(X,Tr)\sim U}[\mathcal{F}_T(\lambda, Tr \cup X) - \mathcal{F}_T(\lambda, X)] \tag{8}$$

where $U$ is the set of users who engaged with the treatment post.

## 3.3 Treatment Effect Calculation

In the above sections, we define the causal structure model and the treatment effect. However, the above formulas are hard to compute. Therefore, in this subsection, we will derive a computable formulation of the treatment effect. We will start from the following theorem:

**Theorem 1.** *For a user $u$, if the intensity function $\lambda(\boldsymbol{f}, t|Tr \cup X)$ is known, then we have:*

$$\mu(t, t+T, \lambda_1, Tr \cup X) = \int_{sup(\boldsymbol{f})} d\boldsymbol{f} \int_t^{t+T} \lambda(\boldsymbol{f}, t|Tr \cup X)dt \tag{9}$$

$$\phi(t, t+T, \lambda_1, Tr \cup X) = \int_{sup(\boldsymbol{f})} \boldsymbol{f}d\boldsymbol{f} \int_t^{t+T} \lambda(\boldsymbol{f}, t|Tr \cup X)dt \tag{10}$$

The first equation can be trivially proved by replacing the $\boldsymbol{F}$ in Equation 25 with the support set. The second one can be proved with the Campbell's Theorem [5]. A detailed proof is provided in the Appendix A.1. The above formulas contain double integral, which is inefficient to compute. To transform the double integral to a single integral, based on a previous work in spatial-temporal point process [7], we have:

$$\lambda(\boldsymbol{f}, t|Tr \cup X) = \lambda(t|Tr \cup X)p(\boldsymbol{f}|t, Tr \cup X) \tag{11}$$

Thus, we can disentangle $\lambda(\boldsymbol{f}, t)$ and respectively model $\lambda(t_i)$ and $p(\boldsymbol{f}|t)$. More importantly, we can simply model $\mu$ as:

$$\mu(t, t+T, \lambda, Tr \cup X) = \int_t^{t+T} \lambda(t|Tr \cup X)dt \int_{sup(\boldsymbol{f})} p(\boldsymbol{f}|t, Tr \cup X)d\boldsymbol{f} = \int_t^{t+T} \lambda(t|Tr \cup X)dt \tag{12}$$

And with this formula, we have:

$$\begin{aligned}
\phi(t, t+T, \lambda, Tr \cup X) &= \int_{sup(\boldsymbol{f})} \int_t^{t+T} \boldsymbol{f}\lambda(t|Tr \cup X)p(\boldsymbol{f}|t, Tr \cup X)d\boldsymbol{f}dt \\
&= \int_t^{t+T} \lambda(t|Tr \cup X)dt \int_{sup(\boldsymbol{f})} \boldsymbol{f}p(\boldsymbol{f}|t, Tr \cup X)d\boldsymbol{f} \\
&= \int_t^{t+T} \lambda(t|Tr \cup X)\mathbb{E}[\boldsymbol{f}|t, Tr \cup X]dt
\end{aligned} \tag{13}$$

The above formulas contain only single integrals. Thus, they can be efficiently approximated with summation: $\int_{x_1}^{x_2} f(x)dx \approx \sum_{i=0}^{(x_2-x_1)/\Delta x} f(x_1 + i\Delta x)\Delta x$

# 4 Neural Estimation of Treatment Effects

The above section construct a causal framework that can measure the impact of a given social media post based on the change of $\lambda(t|Tr \cup X)$ and $p(\boldsymbol{f}|t, Tr \cup X)$. In this section, as shown in Figure 2, we will further discuss how to estimate the impact with a neural temporal point process model.

## 4.1 Learning Conditional Intensity Function via Maximum Likelihood Estimation

The log-likelihood of an observed event $(f, t)$ (no matter an engagement event or an generation event) can be written as:

$$\log p(\boldsymbol{f}, t|Tr \cup X) = \log \lambda(t|Tr \cup X) - \int_{t_n}^t \lambda(t|Tr \cup X)dt + \log p(\boldsymbol{f}|t, Tr \cup X) \tag{14}$$

where $t_n$ is the timestamp of the last event in the set $Tr \cup X$. The above equation provides us with a way to learn $\lambda(t|Tr \cup X)$ and $p(\boldsymbol{f}|t, Tr \cup X)$ by maximizing the likelihood of each event given the historical information (treatment and covariates). To enable the model to make correct prediction for both $\lambda(\boldsymbol{f}, t|Tr \cup X)$ and $\lambda(\boldsymbol{f}, t|X)$, we construct two kinds of samples to train the functions:

**Samples with valid Treatment**: If for a generating event $(\boldsymbol{f}^{(g)}, t^{(g)})$, its most recent previous event is an engagement event $(\boldsymbol{f}^{(e)}, t^{(e)})$ (in other words, the user does not have other activity between the engagement event and the generating event), then we can construct a sample $(Y, Tr \cup X)$, where $Y = (\boldsymbol{f}^{(g)}, t^{(g)})$, $Tr = (\boldsymbol{f}^{(e)}, t^{(e)})$, and $X$ is a sequence that contains all engagement events and generation events before $Tr$.

**Samples without Treatment**: If for a generating event $(\boldsymbol{f}^{(g)}, t^{(g)})$, its most recent previous event is still a generating event (in other words, the user generate two original posts without engaging with other posts), then we can construct a sample $(Y, X)$, where $Y = (\boldsymbol{f}^{(g)}, t^{(g)})$ and $X$ is a sequence that contains all en-

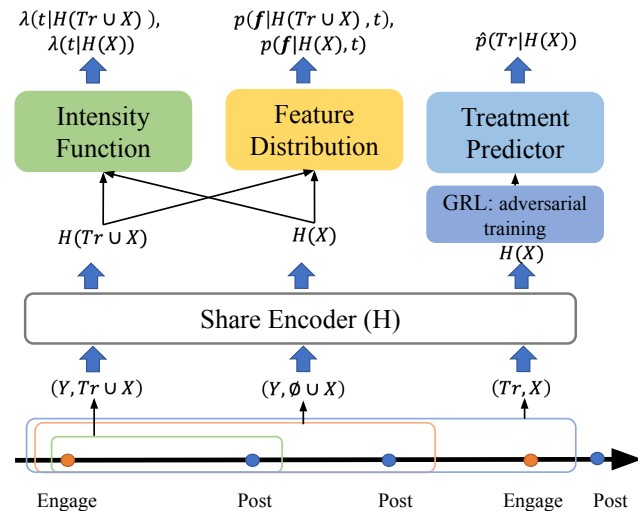

Figure 2: The proposed neural model to estimate the impact of misinformation.

gagement events and generation events before $Y$. In this sample, between the last generation event in $X$ and $Y$, there is no interruption from treatment. Thus, it helps the model to learn $\lambda(t|X)$ and $p(\boldsymbol{f}|t, X)$

For a sample $(Y, Tr \cup X)$ or $(Y, X)$, we first use a shared encoder $H(\cdot)$ to project $(Tr \cup X)$ (or $X$) to a representation vector $h = H(Tr \cup X)$ (or $h = H(X)$ for the case where treatment is NULL). Then we model the intensity function and feature distribution as $\lambda(t|h)$ and $p(\boldsymbol{f}|t, h)$. For $\lambda(t|h)$, following FullyNN, we use a multi-layer perceptron $MLP(h, t)$ to model its integral $\int_{t_n}^{t} \lambda(t|h) dt$. The MLP's partial derivative with respect to $t$ is $\lambda(t|h)$.

To model $p(\boldsymbol{f}|t, h)$, a straightforward solution borrowed from generative deep learning is to apply a neural network, i.e. the decoder, to transform a simple distribution, e.g. a Gaussian distribution whose parameters are decided based on $h$ and $t$, to a complicated distribution. The decoder can be trained via different loss function, like reconstruction error (variational auto encoder) and likelihood (normalizing flow)[3] [44, 7]. However, this method has an important drawback: its conditional expect $\mathbb{E}[\boldsymbol{f}|t, h]$ does not have a closed-form solution. To compute the expect, we can only apply sampling or approximation, e.g. forwarding the expect of the simple distribution into the decoder[4]. To address the above challenge, we propose to explicitly model $p(\boldsymbol{f}|t, h)$ with a mixture of Gaussian distributions:

$$p(\boldsymbol{f}|t, h) = \sum_{j=1}^{m} w_j(t, h) g\left(\frac{\boldsymbol{f}_i - \boldsymbol{u}_j(t, h)}{\sigma_j(t, h)}\right) \tag{15}$$

$$w(t, h) = \text{softmax}(MLP_w(h, t)), \sigma(t, h) = \exp(MLP_\sigma(h, t)), \boldsymbol{\mu}_j(t, h) = MLP_{\boldsymbol{\mu}}^{(j)}(h, t) \tag{16}$$

where $w_j$ is the mixture weight, $\sigma_j$ is a scalar, $\boldsymbol{u}_j$ is a vector with the same dimension as $\boldsymbol{f}$, and $g(\cdot)$ is a standard multivariate gaussian distribution $\mathcal{N}(\boldsymbol{0}, \boldsymbol{I})$ whose covariant matrix is an identical matrix. Although the formula of each component is simple, their mixture has a theoretical guarantee on universal approximation to all distributions [46]. Because the expect of a mixture distribution is the mixture of the expects, we have a closed-form solution for $\mathbb{E}[\boldsymbol{f}|t, h]$:

$$\mathbb{E}_{\boldsymbol{f} \sim p(\boldsymbol{f}|t, h)}[\boldsymbol{f}|t, h] = \sum_{j=1}^{m} w_j(t, h) \boldsymbol{u}_j(t, h) \tag{17}$$

---

[3] GAN is not suitable because for fixed $h$ and $t$ we only have one sample, which can be easily memorized.
[4] Because the decoder $D$ is a non-linear function, $\mathbb{E}[D(x)]$ is usually different from $D(\mathbb{E}[x])$

## 4.2 Adversarial Balanced Neural Temporal Point Process

As discussed in the related work section, by maximizing the likelihood of the posts generated by the users, we can train a neural network that predict $\lambda(t|Tr \cup X)$ and $p(\boldsymbol{f}|t, Tr \cup X)$. However, previous works have proved that if we do not balance the bias from the correlation between treatment and covariates, the model will tend to give biased prediction and thus can not give precise estimation of the treatment effect. A crucial bias in social media data is information cocoon: personalized recommendation systems will deliver user the contents that they are interested in. For example, it will deliver more anti-vaccine posts to anti-vaccine users because they are more likely to be interested in those contents. As a result, the anti-vaccine users will engage more with anti-vaccine posts.

To address the above issue, following previous works in neural counterfactual prediction, we apply domain adversarial training to learn a representation $h$ that is invariant to such a bias[4]. More specifically, we hope to learn a encoder $H$ such that for any two users with different history $X_1$ and $X_2$, $p(Tr|H(X_1)) = p(Tr|H(X_2))$ for the same $Tr$. In other words, in the representation space, the probability that the two users interact with the same post at the same time should be same, which is the same as psychology experiments that divide the experimental and controlled groups randomly. To achieve this object, we apply adversarial training to remove the information about future treatment from the representation of covariates. More specifically, we additionally train a treatment predictor $\hat{p}(Tr|H(X))$ by modeling $\lambda_{tr}(t^{(e)}|h)$ and $p_{tr}(\boldsymbol{f}^{(e)}|t, h)$ with the encoding $h$ of the historical covariates $X$. However, between the treatment predictor and the encoder, we insert a **gradient reversal layer (GRL)**[13, 28, 4] to reverse the sign of the gradient. Thus, when we optimize the treatment predictor to maximize the likelihood of the observed treatment $Tr = (\boldsymbol{f}^{(e)}, t^{(e)})$, the GRL will make the encoder to minimize the likelihood. This process leads to the following min-max game:

$$\min_H \max_{\hat{p}} \mathbb{E}_{Tr,X \sim p(Tr,X)} \log \hat{p}(Tr|H(X)) \tag{18}$$

The following theorem (proof provided in Appendix A.1) provide theoretic guarantee that the above adversarial training reduce the bias introduced by treatment-covariate correlation (e.g. recommendation system and personal interest):

**Theorem 2.** *Given the following min-max game:*

$$min_H max_{\hat{p}} \mathbb{E}_{Tr,X \sim p(Tr,X)} \log \hat{p}(Tr|H(X)) \tag{19}$$

*the min gamer's Nash balanced solution $H^*$, ensures for any $X_1, X_2$, the following equation holds:*

$$p(Tr|H^*(X_1)) = p(Tr|H^*(X_2)) \tag{20}$$

*where $p$ denote the ground-truth conditional distribution of treatment given encoding.*

## 5 Experiments

On real-world social media platforms, the ground truth causal effects of user engagement with posts, no matter misinformation post or information post, are unknown. To address the unknown ground truth causal effect, previous works of causality analysis evaluate their models on synthetic dataset. In this paper, following previous works, we evaluate the performance of our model and compare it with baselines on synthetic dataset. Then we apply our proposed method to evaluate the impact misinformation on a real-world social media data about COVID-19 vaccine collected from Twitter[5].

### 5.1 Experiments on Synthetic Data

**Synthetic Data Generation:** To simulate the real-world social media, we generate 15000 users and 120 post of news. Each user $i$ is represented with a hidden vector $u_i$, which correspond to the status of a social media user. Each piece news $n$ has two randomly generated feature vectors: a topic vector $v_{topic}(n)$ and an inherent influence vector $v_{in}(n)$. Each user has two kinds of activities: (1) engaging with one of the 120 news post and (2) posting a post with original contents. The chance that a user engage with a post is decided by $v_{in}(n)$ and $u_i$, simulating **information cocoons**. Engagement event with news post $n$ will change the hidden status $u$ of the user. The scale and direction of the change are decided by the current user status, the topic vector and the inherent influence vector jointly. For

---

[5]Code and data will be provided in `https://github.com/yizhouzhang97/CNTPP`

Table 1: Estimation Error to the ground-truth ITE

| Method | Accuracy ↑ | RAE ↓ | RRSE ↓ | Decoder Inference Time |
|---|---|---|---|---|
| FullyNN | 73.0% | 0.865 | 0.901 | 7.13ms |
| CNTPP-VAE (Approximation) | 85.9% | 0.279 | 0.503 | **4.05ms** |
| CNTPP-VAE (Sampling) | 87.8% | 0.237 | 0.454 | 29.34ms |
| CNTPP(Ours) | **88.0%** | **0.234** | **0.448** | 7.12ms |

each user, the engagement events and the posting events are modeled through two Hawkes process respectively. Both Hawkes processes are influenced by user status $u$. Also, the feature of each posting event $\boldsymbol{f}$ is drawn from a distribution $P(\boldsymbol{f}|u,t)$ characterized by a random parameterized multi-layer perceptron (MLP) taking $(u,t)$ and random noises as input and output $\boldsymbol{f}$. Thus, the engagement with the news post will have causal effects on the two processes. Since we have all parameters of the model, we can calculate the ground-truth ITE defined in Eq. 7 for the synthetic dataset. The details of the data generation algorithm is included in the Appendix B.2.

**Baselines:** To the best of our knowledge, the causality effect on temporal user behaviour from misinformation is not explored by previous works. Thus, we lack well-established baselines for this specific task. To address this issue, we select some baselines from previous works on temporal point process and temporal causal inference and extend them to adapt our setting. **FullyNN** [25] is a non-causal neural temporal point process that predict the user future behaviours without considering causal effect. We select it because has the same neural architecture as our model. It can also be regarded as our model's variant **w/o adversarial balancing**. **Neural-CIP**[6] is an extension of CIP [14], which aims at discovering causal effect of event pairs in temporal point process. We further compare our model with an ablation variant: **CNTPP-VAE**. It replace our GMM-based decoder with a Variational Auto-Encoder [17]. Since VAE does not have a closed form solution of feature expect, we report the results applying sampling and approximation separately.

In this work, we will evaluate the proposed model in two aspects:

**ITE Estimation**: we will evaluate the model by comparing ITE estimated by the model and the ground-truth ITE. We report three metrics: **Accuracy** (the model need to correctly predict whether the engagement increase or decrease each dimension of the expected average feature), Relative Absolute Error (**RAE**) and Relative Root Square Error (**RRSE**). In addition, we also report the inference time of our model to reflect our model's efficiency.

Table 2: Causal Effect Inference

| Method | MatDis↓ | LinCor↑ |
|---|---|---|
| Neural-CIP | 0.90 | 0.04 |
| FullyNN | 0.93 | 0.236 |
| CNTPP-VAE (Approximation) | 0.84 | 0.303 |
| CNTPP-VAE (Sampling) | **0.76** | 0.287 |
| CNTPP (Ours) | 0.77 | **0.310** |

As shown in Table 1, our model establishes new state-of-the-art on all three quantitative metrics. This means that our model can fully utilize the causal information from the multivariates point process for unbiased treatment effect estimation. In particular, the model with causal analysis outperforms the model with direct neural network prediction (FullyNN), which leads to results that are not causally related. Simultaneously, our model outperforms all baselines in all three metrics of estimation precision. Although CRN-VAE incorporated with sampling method can achieve a performance very close to us, it spends substantially longer inference time.

**Causal Effect Inference**: CIP defines the treatment effect in a way different from our model. Thus ITE estimation experiment is not fair for it. For a fair comparison, and also to further prove that our model can achieve unbiased estimation, we use all the models to predict the ATE of each news post. Then we evaluate the correlation between the learnt ATEs and ground-truth average change of the news post to all users' hidden statuses. The more correlated the learnt ATE is, the better it reflect the inherent causal effect of the engagement on users. To evaluate the correlation, we apply the following two metrics: **MatDis** and **LinCor**. MatDis evaluates the similarity between the ATE-Distance matrix and Hidden-Status-Distance matrix. LinCor evaluates the linear correlation between the learnt ATE

---

[6]Because the authors did not opensource the code of original CIP and one important precious work that is crucial for CIP, we implement a version of CIP that apply neural network rather than graphical causal model.

Table 3: Comparison of (normalized) Average Sum of Distances with different methods on the real-world dataset. This metric reflect how well the a group of data points is clustered.

| Methods | ASD↓ | ASD$_{in}$ ↓ | ASD$_{mis}$ ↓ |
|---|---|---|---|
| Event Feature | 0.123 | 0.123 | 0.122 |
| FullyNN | 0.073 | 0.069 | 0.072 |
| CNTPP (Ours) | **0.045** | **0.042** | **0.044** |

and the ground-truth average hidden status change. Details of the two metrics can be found in Appendix B.4. From Table 2, the ATE of our model best reflect the ground-truth average change of the news post on the simulated data for both two evaluation metrics. This suggests that our model has the potential to discover the influence of misinformation on social media users' hidden status, e.g. interest and idea.

## 5.2 Experiments on Real World Data

In this section, we apply our proposed model on the Twitter dataset to estimate misinformation impact on social media scenario. We apply the data set collected in [49, 33], including a total of 16,9008 tweets with labels from 24,192 users over a 5-month period from 2020/12/09 to 2021/04/24. Notably, we focus on understand how the tweets that users retweeted influence their behavior of posting original tweets. For each post, its feature $f$ includes: text representation (extracted with a pre-trained BERT), sentiment score and subjectivity score. We discover the following two phenomenons with our model.

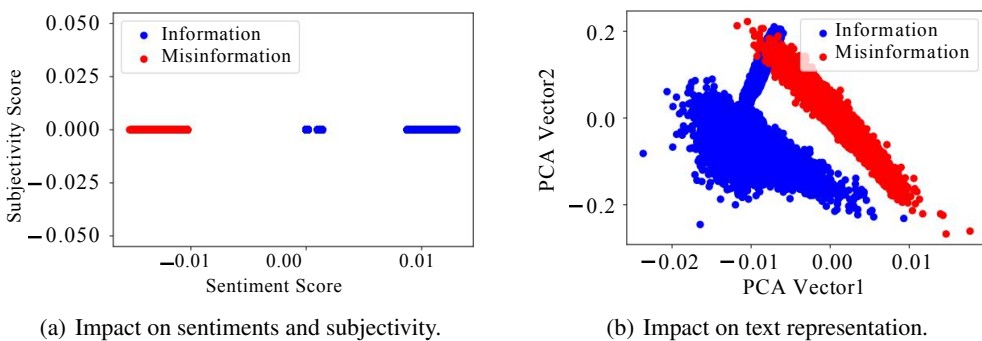

(a) Impact on sentiments and subjectivity.  (b) Impact on text representation.

Figure 3: Analysis on real world social media data

**Identifiability between misinformation and information in influencing people's narratives**: For the desired outcome, we analyze the distinguishability between "retweeting fake news" and "retweeting true news" events. More specifically, for each retweeting event, we use its ITE estimated with our model as feature (dimension reduced via PCA [21]) and whether the content is information or misinformation as label. As shown on Figure 3(b), we can verify the identifiability of our proposed method as the treatment effect of two types of the news are substantially different. This discovery not only supports that misinformation and information influence people's behaviour in different ways, but also provides us with a new paradigm to detect fake news. We also calculate Normalized Averaged Sum of Distances (NASD, details in Appendix B.7) for information cluster, misinformation cluster and their joint set. The lower these metrics are, the better that information and misinformation are distinguished. The comparison of our model against FullyNN and event features on this metric is shown in Table 3. As we can see, the ITE learnt by our model can identify information and misinformation better than the baselines.

**Misinformation is hurting people's subjective emotion related to COVID vaccine**: To understand the influence of misinformation in a more intuitive way, we analyze the impact of retweeting events on users' average sentiment score and subjectivity score in the future. The higher subjectivity the content gets, the more personal opinions rather than factual information text contains. Then, we use the proposed model to generate the estimated ITE for each retweeting event and plot the results

of fake news and real news. As shown on Figure 3(a) (x-axis for sentiment score and y-axis for subjectivity score), we find that both information and misinformation do not substantially influence the users' subjectivity. However, information tends to make people optimistic about vaccines (true news increases the sentiment score), while fake news tends to make people feel negative about vaccines. This discovery strongly supports the hypothesis that misinformation is hurting people's subjective emotion toward COVID-19 vaccines, and suggests that misinformation could be causally responsible for vaccine hesitancy.

## 6 Broader Impact and Limitations

The predicted ITE scores of our model can bring impacts from two perspectives: **misinformation mitigation** and **misinformation research**. First, the predicted ITE scores help platforms allocate resources better for more efficient and effective misinformation mitigation. Here, resources include a wide range of specific concepts, including but limited to the efforts of human verifiers, users' capacity to accept and spread the contents for clarification, and so on. Second, our proposed model provides researchers with a data-driven algorithmic tool to bridge the research in user behavior modeling and misinformation. This tool can help researchers in different ways, e.g. providing researchers a set of potential misinformation factors that could influence user behaviours, understanding misinformation campaigns, which spread misinformation with specific topics or narratives to influence public opinions, and designing better evaluation metrics for fake news detection[7].

The proposed model also has some limitations. First, it mainly focus on the causal effect of engagement on posting. However, in real-world social media, there could be other impacts of misinformation, such as changing the user's preference of engagement, topics of interest and community identity [47]. Also, due to the limitation of synthetic algorithm and the meta data in the real-world data, we did not consider that different types of engagement may have different impact strength. In addition, the real-world dataset experiment only consider one dataset related to COVID-19, which is a single topic dataset. Although our model does not prohibit from being generalized onto multi-topic datasets, e.g., PolitiFact [43] and GossipCop [40, 39, 41], how to verify the model performance and reliability on a multi-topic dataset is still questionable. These limitations provides a strong motivation for further exploration on this paper's topic in future works. Potential directions may include how to verify the model's reliability on multi-topic datasets, how to generate synthetic data with more details and how to model more causal effect in real-world social media.

## 7 Conclusion

In this paper, we propose a framework to describe the causal structure model and causal effect about how misinformation influence online user behaviours. We further design a neural temporal point process model to conduct unbiased estimation on the causal effect in a data-driven approach. Experiments on synthetic dataset verify the effectiveness and efficiency of our model. We further apply our model on real-world dataset from Twitter and recognize identifiable causal effect of misinformation. The experiment results suggests that the misinformation about COVID-19 vaccine is hurting people's subjective attitudes toward vaccines. However, it is also noticeable that our model is a statistical machine learning model. Consequently, all of its estimation can only be regarded as a reference rather than judgement. Also, misinformation campaigns could use the proposed approach to direct their editors to write more impactful fake news. A probable strategy to address this potential problem is to require social media platforms to raise necessary alerts to those suspicious articles that seems to be optimized. We discussed the strategy to detect such articles in the Checklist.

## 8 Acknowledgement and Funding Disclosure

This work is supported by NSF Research Grant IIS-2226087. Views and conclusions are of the authors and should not be interpreted as representing the social policies of the funding agency, or U.S. Government. Yizhou Zhang and Defu Cao are also partly supported by the Annenberg Fellowship of the University of Southern California. We sincerely appreciate the feedback, comments and suggestions from our anonymous reviewers.

---

[7]We are thankful to our anonymous reviewers for the discussion on this issue.

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
