# OpenReview forum: "Counterfactual Neural Temporal Point Process for Estimating Causal Influence of Misinformation on Social Media"
_NeurIPS.cc/2022/Conference — NeurIPS 2022 Accept_

### Official Review · Reviewer_vdd1 · 2022-07-04

**Rating:** 6
**Confidence:** 3
**Soundness:** 3 good
**Presentation:** 3 good
**Contribution:** 3 good

**Summary:**

This paper proposes to estimate the impact of misinformation on real-world social networks. It constructs a causal framework and leverages techniques such as neural temporal point process and gaussian mixture models for this task. Experiments were conducted on synthetic and real-world datasets to demonstrate the proposed approach's effectiveness.

**Questions:**

1. What does ITE stand for? This might be domain-specific jargon and one could infer from line 156 that it denotes "individual treatment effect", while I feel that the authors could explain this at the first occurrence of ITE. This would help people like me to understand your work better :)

2. There are a lot of details regarding the synthetic dataset: "hidden vector $u_i$", "topic vector $v_{topic}(n)$", "influence vector $v_{in}(n)$", "a post of original contents". (line 265-268) Maybe I am missing something, but Section A.3 in the appendix does not seem to describe every single detail of the synthetic dataset. Since the authors only compared the proposed approach with existing methods on the synthetic dataset, these dataset details are crucial to validate that the synthetic dataset is a good indicator of model performance.

3. In checklist 3(c), the authors state that error bars are provided "in the appendix". In Section A.4, the authors do provide the error bars of inference time on the ITE estimation task. However, the error bars of performance metrics, such as Accuracy, RAE, and RRSE in Table 1 as well as MatDis and LinCor in Table 2, are perhaps more important, especially given that the proposed approach only slightly outperforms CRN-VAE (Sampling). I hope the authors could provide the error bars for these evaluation metrics.

4. For the real-world dataset, the authors simply point to [31, 25], while I would like to see more explanation. Why this dataset? Why not existing datasets on fake news detection such as PolitiFact or GossipCop? This real-world dataset seems to only focus on COVID, so maybe a multi-topic misinformation dataset would better support the evaluation?

5. I personally believe that Figure 3(b) could be stronger if the authors also plot this figure for other baselines. Yes, it seems that the estimated ITE vectors could differentiate between real and fake news, but perhaps a language model encoding of news text could achieve the same effect? What about baseline methods in Tables 1 and 2? I would hope for a quantitative comparison (using clustering metrics such as homogeneity score or V-measure) between the proposed approach of baseline methods in the format of Figure 3(b).

6. I am curious about the results in Figure 3(a). It seems that all pieces of news result in almost 0 change in user subjectivity. In light of this, should the subjectivity score remain a factor in the causal framework? I would like to hear about the authors' remarks on this issue.

7. I am excited about the potential of this approach to understanding the impact of misinformation on social media. However, I feel like this paper could better position itself in the context of misinformation research. Specifically, what are the implications of the predicted ITEs towards mitigating online information? How can other researchers use this model to advance misinformation research and address the infodemic?

8. In checklist 1(b), the authors describe this work's limitations as "certain baselines outperform us". I would like to see more discussion about the limitations of the proposed method itself. For example, the framework only models two user activities: post and engage, while there are other user features and there are different types of online engagement such as like and block, which signals vastly different user attitudes. Besides, the method is only evaluated on one real-world dataset that is COVID-related, while whether the approach would generalize to multi-topic real-world social media discourse is questionable.

9. In checklist 1(c) and Section 6, the discussion about potential negative societal impact could be enhanced. For example, operators of misinformation campaigns could use the proposed approach to evaluate how different pieces of misinfo perform in deceiving people and thus design highly misleading misinformation strategies. I would suggest a little more discussion about the societal impact.

**Limitations:**

Please refer to the questions section. I will definitely read the authors' follow-up comments and update my rating accordingly. :)

**Strengths And Weaknesses:**

Strengths:
- proposes a new and interesting direction of misinformation research
- the proposed approach is well motivated and clearly grounded yet improved from previous works
- superior experiment results on both synthetic and real-world datasets

Weaknesses:
- important details about datasets and experiments are missing
- experiments could be improved to better prove the author's claims
- underexplored implications for misinformation research and societal impact

---

> ### Author Response · Authors · 2022-08-02
> **Responses to reviewer vdd1:**
>
> Dear reviewer vdd1:
>
> We are sincerely grateful for your questions and comments. After reading your review, we are so excited that you would like to spend so much time and effort on reading our paper. The following are the answers to your questions (question 1-5 is in this comment, and question 6-7 will be addressed in a separate comment due to the comment length limit):
>
> 1. Thank you for your comments! We do mention Individual Treatment Effect in the main content yet did not link it directly to ITE. We have updated our draft accordingly.
>
> 2. In the updated appendix, we have added further details and a figure to help readers understand the details better. After acceptance, we will add those important information accordingly.
>
> 3. All the models that we present in the papers, except CRN-VAE(Sampling), will give deterministic estimation to ITE once they are well trained. Thus, the error bars of effectiveness metrics such as accuracy and RRSE, are only valid for CRN-VAE(Sampling), and can not be compared with other models.
>
> 4. Thank you for this question. The answer to this question could be long. In the very beginning, our motivation on this topic is to answer a question that was first proposed yet not resolved by another paper [1]. The question is: how much could misinformation influence people’s attitudes toward vaccines? Then, we generalize our objective to a more impactful target: developing a data-driven machine learning model that can help both platforms and researchers in the social media analysis area to recognize the causal effect of a piece of misinformation on user behaviors. So naturally, after evaluating the proposed work on our synthetic data, we further apply it on a vaccine dataset in [1] and recognize identifiable differences between misinformation and information. Also, in the area of social media analysis, researchers focus more on understanding people’s reaction and attitudes toward recent public events. Therefore, from the perspective of making more impacts and more contributions on the sub-area of social media analysis, we select the recent dataset about vaccines. Meanwhile, since our model is a data-driven deep learning model, it can also be applied on datasets about other topics. As for generalizing this model to multi-topic datasets, our model design and theoretical analysis are not specific to single-topic scenarios. Therefore, we believe that our model can be applied on multi-topic datasets to learn ITE scores. A probable issue of multi-topic dataset is that the ground-truth Average Treatment Effects (ATE) may not be so recognizable in multi-topic scenarios. For example, when we only discuss the misinformation about COVID-19 vaccines, most of the fake news is trying to convince people that COVID-19 vaccines are unsafe and ineffective by twisting the news about side effects. While in information posts, the ratio of such posts is much less. Thus, it is natural that misinformation averagely decreases people’s sentiment scores, compared to information. However, if we talk about other topics, then the effects could be inverted. Under other topics, the misinformation might make people over optimistic toward the pandemic. If we mix the data from different topics, then the ground-truth ATE may not be recognizable. Although in such a scenario, our estimation toward ITE is not expected to be influenced, we do not have ground-truth ITE scores to evaluate the estimation toward the ITE scores.
>
> 5. Thank you for your suggestions. In the updated appendix for rebuttal, we added the virtualization of the ITE scores estimated with the FullyNN baseline and the event features (the BERT embedding with PCA reduction). We also compute Normalized Average Sum Distance (the lower the metric is, the better the data is clustered) for them and our model. Our results significantly outperform the baselines.

---

> > ### Author Response · Authors · 2022-08-02
> > **Continue**
> >
> > 6. This question is very valuable! We choose to remain subjectivity score in our experiment framework for two reasons. First, we try to verify an important ability of our model: based on the observed data our model can suggest whether one factor (e.g. engagement with misinformation of specific topics like COVID-19 vaccine) could causally influence the other factor (e.g. subjectivity and sentiment) or not. Figure 3(a) reflected that the model suggests that the subjectivity is not causally influenced by social media engagement as substantially as sentiment scores. This determination is consistent with our prior intuition: emotions are far more easily influenced by social media engagement than subjectivity. Such an ability of determination is important in practice because sometimes our prior knowledge can not tell us which factors are causally influenced and which are not. In such a case, we can simply put all potential factors in the data and ask the model to determine based on the data. Second, although subjectivity score as an outcome is not substantially influenced by misinformation engagement, it could be useful confounders or covariates for the model to determine the individual treatment effect.
> >
> > 7. Thank you for this helpful comment! We will present the benefits brought by the predicted ITE scores from two perspectives (misinformation mitigation and misinformation research). First, predicted ITE scores help platforms allocate resources better for more efficient and effective misinformation mitigation. Here, resources include a wide range of specific concepts. For example, the time and efforts of human verifiers and experts is an important resource. Because current fake detection algorithms are usually based on statistical machine learning and crowd intelligence, the detection results may contain false positives and false negatives. Thus, the platforms have to arrange human verifiers to do the final check and then mitigate the misinformation. Also, users’ capacity to the contents for clarification is another kind of resource for misinformation mitigation. For example, if a user has already viewed a lot of posts containing misinformation and the platform wants to recommend some contents for clarification, a good strategy for the platforms is to recommend the contents for the posts with highest ITE scores for this user rather than recommend all the contents. Besides, similar to online advertisement and influence maximization on social networks, sometimes we will expect the users to help us spread some contents for misinformation mitigation. In such a scenario, the users’ capacity to retweet the contents for clarification is also a kind of resource.
> > Our proposed model provides researchers with a data-driven algorithmic tool to bridge the research in user behavior modeling and misinformation. This tool can help researchers in different ways. First, as we mentioned earlier in the answer to question 6, our model can “suggest whether one factor (e.g. engagement with misinformation of specific topics like COVID-19 vaccine) could causally influence the other factor (e.g. subjectivity and sentiment) or not”. Such an ability can help researchers quickly recognize potential hypotheses, which wait to be validated via further research, survey, analysis and even psychological experiments, from data. Second, this tool has the potential to help researchers better understand misinformation campaigns, which spread misinformation with specific topics or narratives to influence public opinions. The activities of such campaigns are naturally motivated by the purpose to enlarge the impact, or in other words the treatment effects, on users’ opinion from the misinformation spread by them. Thus, our proposed model could be an efficient tool in the research about misinformation campaigns. Third, we can design better evaluation metrics for fake news detection. For example, with our model, we can categorize the fake news in the test sets based on their ITE scores and evaluate the model on them respectively. Such a metric can improve the depth of the current fake news detection models and help us understand them better.
> >
> > 8. We are sincerely grateful for the reviewer to point out some limitations that we did not realize before. For different types of engagement (like comment or retweet), we can add the type information into the event feature f if the metadata contains such information. We have added discussions to them in the checklist. For other limitations, if this paper is accepted, we will have one more page for the main content and thus can add the discussion into it.

---

> > > ### Author Response · Authors · 2022-08-02
> > > **Continue**
> > >
> > >
> > > 9. Thank you for pointing out these potential impacts. Same as question 8, we have added them in the checklist. In the updated draft, we not only mentioned this potential problem, but also discussed possible methods to detect such deceiving articles. A probable strategy to address this potential problem is to require social media platforms to raise necessary alerts to those suspicious articles that seems to be optimized under this model for raising impacts. Such a detection task has multiple potential solutions. One solution is to use anomaly detection models to detect articles with extremely high ITE or ATE scores. Another probable solution is based on gradient and optimization. The process of guiding editors with the proposed model is equivalent to generating texts under a regularization to maximize the ITE or ATE scores estimated by the proposed model. Thus, the "optimized" articles will be closed to an optimal point, where gradients of the ITE or ATE scores with respect to the article feature is statistically lower than other points. Thus, the platforms can detect such articles by computing the article embedding's distance to the closest local optimal point or its gradient scale. If this paper is accepted, we will have one more page for the main content and thus can add the discussion into the paper.
> > >
> > > We are sincerely thankful for you suggestions. We are looking forward to your feedbacks and having more discussions!
> > >
> > > Best,
> > >
> > > Authors of this submission

---

> > > > ### Comment · Reviewer_vdd1 · 2022-08-09
> > > > **Thank you for your response.**
> > > >
> > > > Thank you for your detailed response. All my concerns are adequately addressed and I have updated the score to reflect that. Estimating the influence of online misinformation is a very interesting topic and has also come to my mind in the past, but I never started working on it. I wish the authors all the best in their future work on this topic. :)

---

> > > > > ### Author Response · Authors · 2022-08-09
> > > > > **Thank you for your appreciation!**
> > > > >
> > > > > Thank you so much! It is our pleasure to have the feedback and help from you. We are really fortunate to have a reviewer who have similar ideas and interests on this research topic. We will keep moving forward on this topic, with your encouragement.
> > > > >
> > > > > Best,
> > > > >
> > > > > Authors of this submission

---

> ### Comment · Area_Chair_umZ7 · 2022-08-08
> **Please read and respond to the author response**
>
> vdd1, the authors have provided a detailed response to your review — including answers to your questions. Can you please read and respond to this?

---

> > ### Comment · Reviewer_vdd1 · 2022-08-09
> > **I'm sorry.**
> >
> > Sorry for not responding earlier. I was on vacation last week and only got back yesterday.

---

### Official Review · Reviewer_SxLq · 2022-07-10

**Rating:** 6
**Confidence:** 4
**Soundness:** 3 good
**Presentation:** 3 good
**Contribution:** 3 good

**Summary:**

The paper describes a novel method of tackling the impact of misinformation in social media through the lens of causal analysis via a neural temporal point process model to estimate the effects using generated synthetic data and real-world data in the domain of healthcare (re: COVID-19 vaccines). The proposed method draws motivation from how misinformation is spread on social media sites and how related previous works have only focused on correlational studies and randomized trials without exploring causality-based studies. To properly evaluate the performance of the causal inference model, the authors used synthetic data. Results show that the proposed method slightly outperforms neural network models without causal modeling and autoencoder-based methods in accuracy and inference time. In the real-world data experiment, the authors draw summarized observations on how sentiments affect the behavior of users in spreading (mis)information and differentiation from the two using estimated ITE.

**Questions:**

1. Is the BERT embeddings concatenated together with the subjectivity score and sentiment score? How were the latter two obtained? Is there a reliability rate for these scores?
2. In Section 5, there is a mention that previous models used synthetic datasets for evaluating proposals on causality analysis. Just out of plain curiosity, were the authors able to retrieve the generation parameters from previous works and apply the synthetic data generation close if not equal to the parameters used for potential comparison? Discussion on this is encouraged.
3. What do/does the author/s infer from the overlap of (mis)information based on ITE scores in Figure 3B?
4. In the experiment setup with both synthetic and realistic data, can the engagement actions (like, retweet, comment) be ranked or weighted according to how likely can trigger the user to spread (mis)information? For example, a user may have a higher chance of spreading a misinformed post after liking it *without* commenting on it as we don’t always assume that both actions are done together all the time. In this case, the *like* has more *weight* than the comment.

**Limitations:**

The paper poses no serious negative societal impact as long as the results, as mentioned by the authors, are only taken into consideration as additional reference and not an assumption of absolute truth in a decision-making scenario. In addition, as mentioned in the sections above, I suggest the authors include a Limitations section to cover discussion on the limitations of the study including the breadth of the domain used for measuring the impact and effect of (mis)information, data generation settings, and evaluation.

**Strengths And Weaknesses:**

I appreciate the paper’s sufficient and balanced level of technicality on treatment effects, adversarial training, and the proposed causal structure model. I find the motivation for applying causal analysis in alleviating fake news on social media solid. I also commend the authors for structuring the paper in such a way that it’s very easy to follow, read, and understand. I have a few points, however, that can be considered and discussed to further improve the quality of the work:

1. The dataset used (both real-world and synthetic) should be at least described sufficiently in the paper including mentions of limitations of the domain used (health-related topics only), language specifications, and search parameters such as geolocation used to name a few. The same way goes for the evaluation metrics used: MatDis and LinCor. I believe these are essential information that should be present in the paper itself without being directed to the Appendix. In case of acceptance, I highly recommend including this information for completeness.
2. Discussion on comparison between the proposed method vs. the CRN-VAE in Section 5.1 for causal effect inference and estimation residuals can be improved to add more depth. What is it with the sampling method of CRN-VAE that makes it relatively at par with the proposed method but lags behind inference time? Moreover, the note on heavily revising CRN-VAE confuses the reader. Aside from getting the sampled and approximated outputs, what other changes were made to the framework?
3. If there are no statistical tests done, I suggest switching the term significantly to substantially as the word is already overused in literature without support.

---

> ### Author Response · Authors · 2022-08-02
> **Responses to reviewer SxLq**
>
> Dear reviewer SxLq:
>
> We are greatly thankful for your appreciation and suggestions. It is our pleasure to have your help on improving the quality of our work. We have addressed the wording issue (change significant to substantial) and discussed the limitations and potential negative impact in the checklist due to the page limits. If this paper gets accepted, we will get one more page to put the discussions into the main content. The following are the answers to your questions:
>
> 1. What is it with the sampling method of CRN-VAE? In CRN-VAE, an encoder will project the covariate and treatment to a multi-dimensional Gaussian distribution. Then a hidden vector will be sampled from this distribution and forward into a decoder to predict the future event feature f. To estimate our proposed ITE scores, we need to infer the feature expect E(f). However, since in CRN-VAE the decoder is a neural network (more detaily, a Multi-Layer Perceptron), we can not acquire a closed-form solution, unlike our proposed model. Thus, we have two strategies: (1) approximation, which directly forward the mean of the multi-dimensional Gaussian distribution into the decoder, and (2) we sample multiple hidden vectors independently from the multi-dimensional Gaussian distribution and forward all of them into the decoder and calculate their means. The time complexity of the sampling strategy will be O(n), where n is the number of samples. However, for our method and the approximation method, we only use the whole model once so the time complexity will be O(1).
>
> 2. What revisions are made to CRN to acquire CRN-VAE? Original CRN is a model for time series with fixed time gap between two events (observation to patient in their original paper). Their encoder is a recurrent neural network (RNN, same as us) and decoder contains two networks predicting future treatment type (a classifier) and future outcomes (a regression model) respectively. Similar to our model, they insert a Gradient Reversal Layer in front of the treatment predictor to apply adversarial balance. Because the time gap in their setting is fixed, CRN does not need to predict future event timestamps. To adapt CRN to our setting, we heavily revise their decoders. For both treatment predictor and outcome regression model, we disentangle the predictions of future event time (λ(t|Tr,X)) and feature (p(f|Tr, X)) based on the Eq. 14 in our paper. For the intensity function λ(t|Tr,X), same as our model, we apply the FullyNN architecture. For the feature prediction, we use variational auto-encoders as we illustrated in the previous paragraph. After such heavy revisions to the decoder end, the new model CRN-VAE can be regarded as an ablation variant of our model where the GMM model is replaced with VAE when we use the same encoder (RNN). This is the reason why the accuracy performance of CRN-VAE (sampling) is close to our model yet its time complexity is higher: with enough sampling size, which consumes much more time than GMM, the VAE can also estimate the mean without bias.
>
> 3. Is the BERT embeddings concatenated together with the subjectivity score and sentiment score? How were the latter two obtained? No, we separately trained two models. One model takes the two scores as input. The other one uses BERT embedding as input.
>
> 4. Is there a reliability rate for these scores? We used textblob, a widely-used general-purpose sentiment analysis package in Python. On our dataset, since we do not have the ground-truth of the subjectivity score and sentiment score, we can not compute accuracy. However, a recent paper [1] reported that on a COVID-19 vaccination related dataset (not ours) textblob outperformed other general-purpose sentiment analysis tools (VADER and AFINN). Although this paper also proposed a model that is reported to have superior accuracy, we did not use it as it is not open source. Besides, the proposed model is a deep learning model, whose performance may heavily rely on the distribution gap between their training dataset and our dataset.
>
> 5. Synthetic data in previous works. The previous works we discussed for counterfactual analysis like CRN mainly focus on the medical and healthcare domain (mentioned in line 37), thus the domain is different from ours. In the social media analysis domain, relevant works mainly focus on correlation analysis, which can be directly evaluated on real-world dataset. This is actually one of our contributions: to the best of our knowledge, our model is the first data-driven effort trying to analyze the causal impact of misinformation on user behaviors. Although directly borrowing parameters seems to be challenging, borrowing some ideas from the synthetic algorithms from the healthcare domain could be promising. This idea, although already beyond the main topic of this paper, deserves further exploration in the future work. We also hope to have a further discussion with you on this idea.

---

> > ### Author Response · Authors · 2022-08-02
> > **Continue**
> >
> > 6. What do/does the author/s infer from the overlap of (mis)information based on ITE scores in Figure 3B? There are third possible explanations for the overlap. The first explanation is that, although the average impacts of misinformation and information are very different, sometimes individually a piece of information can cause a similar impact on users as fake news. For example, both true news that objectively report the potential side effects of vaccines and fake news that slightly twist facts can hurt people’s confidence in vaccines. The second explanation is that they are caused by the gap between our model’s prediction and the ground-truth ITE scores. The third explanation is that it is caused by the dimension reduction algorithm. Because on real-world dataset, we do not have the groundtruth ITE scores, we can not decide which explanation holds, or maybe the overlap is caused by the three factors jointly.
> >
> > 7. Your suggestion is very reasonable. And our model is compatible with the engagement type information: we can just add the engagement type into the event feature. We did not apply this in the synthetic data because we lack prior knowledge about how to set the weights of different engagement types. As for the real-world dataset, its meta data only has the comments and retweets, without “liking” information.
> >
> > We are sincerely thankful for you suggestions. We hope to hear more from you and have more discussions!
> >
> > Best,
> >
> > Authors of this submission

---

> > > ### Comment · Reviewer_SxLq · 2022-08-07
> > > **Acknowledge authors' response**
> > >
> > > This is to acknowledge that I have read the authors' response. So far, my questions about the paper have been sufficiently answered and I thank the authors for taking the time to provide very detailed answers.
> > >
> > > To the authors, in case of acceptance, I would like to suggest that you include the important information you mentioned in responses #2, #6, and #7 in the main paper itself (not in the appendix) as these cover aspects of the variation of CRN-VAE you used, deeper interpretation of the misinformation overlap, and instances of interaction variation in social media (likes, retweets, comments, etc).

---

> > > > ### Author Response · Authors · 2022-08-09
> > > > **Thank you for your appreciation!**
> > > >
> > > > Thank you so much for your feedback and appreciation on our efforts! In case of acceptance, we will definitely include the information since one more page will be given.
> > > >
> > > > Best,
> > > >
> > > > Authors of this submission

---

### Official Review · Reviewer_5Zb3 · 2022-07-12

**Rating:** 4
**Confidence:** 4
**Soundness:** 3 good
**Presentation:** 2 fair
**Contribution:** 2 fair

**Summary:**

This paper proposes a model for the causal structure of the misinformation influence on users. It proposes one of the first studies of this kind building on temporal point processes and modelling users' engagement and posting behaviour as events while misinformation is considered as treatment. A treatment effect evaluation is proposed which projects the intensity function to a vector to measure the individual treatment effect. This is then calculated more efficiently through reformulations and then estimated with a neural temporal point process model. In the experiments, synthetic data is used to show the accuracy of the model, whereas a real-world covid-vaccine Twitter dataset is discussed as a case study to show how misinformation influences users' behaviour differently compared to information, among other findings.

**Questions:**

Can you explain the main intuition of how this model works?

What is the problem definition?

Can you explain why the subjectivity score is always zero in fig 3a? what is the accuracy of the misinformation labels used in 5.2. as well as the sentiment and subjectivity scores? have you done any evaluation on how reliable these are? what is the range for sentiment scores?

Can you explain how your data is different from the references cited? the data sizes do not match.

**Limitations:**

Not discussed in the paper in depth.

Checklist 4d -> using Twitter data is common in research but still, the question of consent is relevant and can be discussed.


**Strengths And Weaknesses:**

Strengths:
- well-motivated paper, and interesting problem
- paper is generally easy to follow and well structured

Weaknesses:
- presentation and writing needs improvement
- many key details are given only in the appendix and the paper is hard to understand as is
- results are not convincing, and hard to interpret


Minor comments:
"This is not applicable for social media post because most meaningful features of the posts, such as geographical information, sentiment score and subjective score, are naturally continuous. Forcibly discretizing them will lose meaningful information." =>  geographical information usually is at the city level, sentiment is often a classification. I am not sure what is the subjective score but overall the argument is weak.

What is S_{in} in line 122?

Many language/grammar errors, examples:

"Consequently, an effective and efficient automatic methodology to estimate the impact of the misinformation on user beliefs and activities."

"to study the relation between misinformation on people’s behaviours."

"Unless getting heavily revision, such previous work can not be simply transferred to our problem scenario."

"the treatment effect can be trivially computed by compare the difference of the outcomes from real world and counterfactual world"

"we are lack of well-established baselines for this specific task."

---

> ### Author Response · Authors · 2022-08-02
> **Response to reviewer 5Zb3**
>
> Dear reviewer 5Zb3
>
> Thank you for reading our paper and finding out some typos. We have corrected them. Below are our responses to your questions:
>
> 1. As we discussed from line 43 to line 66, the main intuition why our model can estimate the ITEs more accurately is that we apply adversarial balance to reweight the samples in the embedding space to simulate the randomized controlled trials in the real world. The reason why our model can reflect the hidden status change of the users is that our model does not require the feature of the events to be categorized.
>
> 3. As we mentioned in Section 3.1, we are trying to define and estimate the impact of the event that a user engaged with a post on the user’s future activities on social media. More specifically, we consider two temporal point processes: the process that a user engages with existing news and the process that a user posts original contents. We are trying to construct a causal framework to formulate the impact of an engagement event on a user’s posting process and then design a data-driven model to estimate such impact.
>
> 3. It is not that the subjectivity score is 0. Figure 3a is showing that the influence of both information and misinformation on a user's subjectivity score are almost 0. The misinformation labels in 5.2 are provided by the dataset itself, not from our models. The range of sentiment score is [-1,1], where 1 means extremely positive and -1 means extremely negative.
>
> 4. In the original dataset, not all posts contain labels of “Misinformation” and “Information”. Thus, we only use those data with labels of “Misinformation” and “Information”, as we mentioned in line 73-74 in the Appendix.
>
> 5. Limitations "not discussed in the paper in depth." We have updated the draft and appendix according to the suggestions from reviwers on the limitation discussion.
>
> 6. "Checklist 4d -> using Twitter data is common in research but still, the question of consent is relevant and can be discussed." We collect the data via academic API, which is provided by Twitter officially.
>
> Thank you for your comments. If you have any other questions, we would like to response further.
>
> Best,
>
> Authors of this submission

---

> > ### Comment · Reviewer_5Zb3 · 2022-08-10
> > **Confirming the response**
> >
> > Thank you for updates in the paper and answering the questions. The new results in the appendix are very relevant and helpful. I have tried reading the paper again and I (as someone working on related topics) am still not able to understand many of the key points and follow along properly. I think the paper addresses an important problem and the results are interesting but it can be written more clearly and results presented more strongly, especially given the strong conclusions provided on the real world data. Beyond making the paper more clear, one suggestion is to do a small manual coding of the sentiment and subjectivity to check the accuracy of the textblob. This will help making the results more convincing.

---

> > > ### Author Response · Authors · 2022-08-10
> > > **Thanks for the response!**
> > >
> > > We value your comments and hopefully we can be more specific in discussing the key points that are confusing to you, which we believe would be definitely helpful for our paper!
> > >
> > > We used textblob, a widely-used general-purpose sentiment analysis package in Python. Since we do not have the ground-truth of the subjectivity score and sentiment score on our dataset, we can not compute accuracy. However, a recent paper [1] reported that on a COVID-19 vaccination related dataset (not ours) textblob outperformed other general-purpose sentiment analysis tools (VADER and AFINN). Although this paper also proposed a model that is reported to have superior accuracy, we did not use it as it is not open source. Besides, the proposed model is a deep learning model, whose performance may heavily rely on the distribution gap between their training dataset and our dataset.
> > >
> > > [1] Reshi, A.A.; Rustam, F.; Aljedaani, W.; Shafi, S.; Alhossan, A.; Alrabiah, Z.; Ahmad, A.; Alsuwailem, H.; Almangour, T.A.; Alshammari, M.A.; Lee, E.; Ashraf, I. COVID-19 Vaccination-Related Sentiments Analysis: A Case Study Using Worldwide Twitter Dataset. Healthcare 2022, 10, 411.
> > >
> > > In fact, we have a complete internal analysis of the textblob, including the Twitter posting sentiment for different vaccines on different dates, which we will also make it publish after this paper accepted. Qualitatively speaking, the textblob can adequately reflect the impact brought by the news event at that time. Meanwhile, we are working on human evaluation and will report the Accuracy according to human knowledge in the final version.
> > >
> > > Hopefully, all the above works will make this paper more convincing to you!

---

### Author Response · Authors · 2022-08-02
**Updated draft and appendix**

Dear reviewers,

Thank you so much for your time and efforts. We have udpated our draft and appendix according to your suggestions. The main revisions (labeled with blue text) includes:

1. Add more discussions to the limitations and potential negative social impact (in checklist). If this paper is accepted, we will have one more page to add this part into the main content.

2. Add more details of the generation of the synthetic data in Appendix (Figure 1). If this paper is accepted, we will have one more page to add this part into the main content.

3. Add experiment results (in Appendix, Table 2 and Figure 2) of some baselines on the real-world dataset and further compars them with our model.

We sincerely hope to hear more suggestions or feedbacks. Looking forward to further discussions!

Best,

Authors of this submission

---

### Public Comment · ~Aniq_Ur_Rahman1 · 2025-01-28
**Data Availablity**

Hello,

Thanks for releasing the code publicly.
Could you please share the specifications of the data after unpacking the pickle file, or provide a sample data file.

Thank you.

---

### Meta-Review · Area_Chair_umZ7 · 2022-08-23

**Recommendation:** Accept
**Confidence:** Less certain

**Metareview:**

This work addresses the interesting problem of measuring the causal effect of exposure to misinformation from observational data culled from social media. Reviewers agreed on the importance and timeliness of this problem. The motivation for causal analysis here is also readily apparent (SxLq), and the proposed point process model seems novel (at least in its application here) and appropriate.

The main weaknesses raised concerned presentation issues (5Zb3,vdd1) and some concerns about experimental details and setup (vdd1,SxLq). The former constitute relatively minor issues which can be readily addressed in revision, and the latter issues were mostly satisfactorily addressed during the response period. Reviewer 5Zb3 raises a concern about the sentiment model used as part of the evaluation, which the authors should discuss in future iterations of the work.

Overall, while the evaluation suggests only modest empirical gains over baselines (which were themselves introduced in this work, for want of alternative existing methods), the work is novel in its investigation of causal methods for understanding the potential influence of misinformation on social media.




**Award:**

No

---

### Decision · Program_Chairs · 2022-09-14

Accept